# Transcriptome Analysis of *Meloidogyne javanica* and the Role of a C-Type Lectin in Parasitism

**DOI:** 10.3390/plants13050730

**Published:** 2024-03-04

**Authors:** Wenwei Chi, Lili Hu, Zhiwen Li, Borong Lin, Kan Zhuo, Jinling Liao

**Affiliations:** 1Laboratory of Plant Nematology, South China Agricultural University, Guangzhou 510642, China; wenwei.chi@sinomdgene.com (W.C.); lizhiwen2017@126.com (Z.L.); baironglin@126.com (B.L.); zhuokan@scau.edu.cn (K.Z.); 2Guangdong Provincial Key Laboratory of Silviculture, Protection and Utilization, Guangdong Academy of Forestry, Guangzhou 510520, China; hulili0113@163.com; 3Guangdong Eco-Engineering Polytechnic, Guangzhou 510520, China

**Keywords:** *Meloidogyne javanica*, transcriptome, C-type lectin, plant immune response

## Abstract

*Meloidogyne javanica* is one of the most widespread and economically important sedentary endoparasites. In this study, a comparative transcriptome analysis of *M. javanica* between pre-parasitic second-stage juveniles (Pre-J2) and parasitic juveniles (Par-J3/J4) was conducted. A total of 48,698 unigenes were obtained, of which 18,826 genes showed significant differences in expression (*p* < 0.05). In the differentially expressed genes (DEGs) from transcriptome data at Par-J3/J4 and Pre-J2, a large number of unigenes were annotated to the C-type lectin (CTL, *Mg01965*), the cathepsin L-like protease (*Mi-cpl-1*), the venom allergen-like protein (*Mi-mps-1*), *Map-1* and the cellulase (endo-β-1,4-glucanase). Among seven types of lectins found in the DEGs, there were 10 CTLs. The regulatory roles of *Mj-CTL-1*, *Mj-CTL-2* and *Mj-CTL-3* in plant immune responses involved in the parasitism of *M. javanica* were investigated. The results revealed that *Mj-CTL-2* could suppress programmed cell death (PCD) triggered by Gpa2/RBP-1 and inhibit the flg22-stimulated ROS burst. In situ hybridization and developmental expression analyses showed that *Mj-CTL-2* was specifically expressed in the subventral gland of *M. javanica*, and its expression was up-regulated at Pre-J2 of the nematode. In addition, *in planta* silencing of *Mj-CTL-2* substantially increased the plant resistance to *M. javanica*. Moreover, yeast co-transformation and bimolecular fluorescence complementation assay showed that *Mj-CTL-2* specifically interacted with the *Solanum lycopersicum* catalase, *SlCAT2*. It was demonstrated that *M. javanica* could suppress the innate immunity of plants through the peroxide system, thereby promoting parasitism.

## 1. Introduction

Root-knot nematodes (RKNs), known as obligate biotrophic plant parasites, are one of the main plant-parasitic nematodes, infecting more than 5000 plant species [1]. They attack the roots of various plants and absorb essential nutrients from highly specialized, multinucleate feeding cells. The genus of RKNs is dominated by *Meloidogyne incognita*, *M. arenaria*, *M. hapla* and *M. javanica*, causing crop losses amounting to hundreds of billions of US dollars each year [2]. *M. javanica* is widely distributed in southern China, affecting many economically important crops, ornamental plants and fruit trees [3].

With the rapid development of biotechnologies, the whole genome sequencing of *M. incognita* and *M. hapla* was completed in 2008 [4,5], which facilitates the molecular function study of effectors in RKNs. Transcriptome and genome sequencing will provide more information for elucidating gene function and provide insights into the interaction between nematodes and hosts. There have been relevant transcriptome studies on *M. enterolobii*, *M. graminicola* and *M. incognita* [6], and in this study the comparative transcriptome sequencing data of *M. javanica* between pre-parasitic second-stage juveniles (Pre-J2) and 14-day post-infection (dpi) parasitic juveniles (Par-J3/J4) were obtained for the first time. Additionality, among the obtained differentially expressed genes (DEGs), 10 C-type lectins (CTLs) were selected as candidates for further investigation.

Lectins, present in plants, animals and microorganisms, are able to specifically bind to monosaccharides or oligosaccharides, which enable them to bind to soluble carbohydrates, or sugar residues of glycoproteins, thus triggering a series of downstream cascade reactions [7]. Lectins could sense the host or suppress host immunity to promote parasitisms of themselves. Recognition between pahtogens and a host is the prerequisite for successful infection of pathogens [8]. This process is affected by many factors, but lectins on the surface of pathogens and glycans on the surface of host cells have been proven to play a critical role [9]. The first characterized lectin was isolated from the seeds of the castor bean (*Ricinus communis*) and took the name ricin [10]. A hepatic lectin was discovered in rabbit liver in 1974 and may be the first lectin of the mammalian origin [10].

There are several types of lectin domains present in nematodes, including the GH18 domain of class V chitinase, hevein domain, legume lectin-like domain, LysM domain, F-type lectin domain, Ricin toxin B chain (RTB) lectin domain, calreticulin domain, M-type lectin domain, CTL domain (CTLD) and galectin domain, of which CTLD and the galectin domain are the most studied [10]. A CTL is a Ca^2+^-dependent glycan-binding protein (GBP) that shares primary and secondary structural homology in its carbohydrate-recognition domain (CRD). The CRD of CTLs is generally regarded as the CTLD, representing a ligand binding motif that binds to sugars, proteins, lipids and even inorganic ligands [11]. One of the most striking characteristics of CTLs is the “WIGL” motif, which is highly conserved in CTLs. A CTL performs its function in the cell adhesion, glycoprotein clearance and innate immunity [10,12]. According to previous studies, CTLs are widely present in free-living nematodes, animal-parasitic nematodes and plant-parasitic nematodes. However, the quantities of CTLs were relatively larger in free-living nematodes compared to plant- and animal-parasitic nematodes. The gene SCN1018, cloned from *Heterodera glycines*, encodes a CTL [13], and soaking *H. glycines* at Pre-J2 in double-stranded RNA of SCN1018 to induce gene knockdown via RNA interference reduces parasitic nematodes in host plants [14]. Eleven CTLs were identified in *Rotylenchulus reniformis* [15]. The CTL Mg01965 from *M. graminicola* can suppress plant defense and promote nematode parasitism [12]. The MiCTL1a from *M. incognita* could suppress the flg22-stimulated reactive oxygen species (ROS) burst through interactions with catalases in *Arabidopsis thaliana* [16].

In this study, a comparative transcriptome analysis between Pre-J2 and Par-J3/J4 of *M. javanica* was conducted, through which a CTL was identified from the obtained list of DEGs and the expression pattern of the CTL was elucidated. The results confirmed that *Mj-CTL-2* could inhibit plant immune responses and play a role in nematode parasitism via interactions with the *Solanum lycopersicum* catalase SICAT2.

## 2. Results

### 2.1. Overview of M. javanica Transcriptome Sequencing, Assembly and Annotations

Illumina Hiseq 2000 was utilized for transcriptome sequencing of *M. javanica* at Pre-J2 and Par-J3/J4. This sequencing yielded 55,759,032 reads from the Par-J3/J4 library and 92,886,548 reads from the Pre-J2 library, with an average length of 257 bp and 261 bp, respectively (Appendix A). There were 34,047 and 45,352 unigenes in the transcriptomes at Pre-J2 and Par-J3/J4, respectively, and a total of 48,698 unigenes were finally obtained after mixed splicing.

In the transcriptome data, 27,544 and 28,706 genes (FPKM > 1) were expressed at Par-J3/J4 and Pre-J2, respectively. Among them, 22,743 genes were expressed at both Par-J3/J4 and Pre-J2, but 4801 genes were expressed only at Par-J3/J4, while 5963 genes were expressed only at Pre-J2 (Figure 1). In total, 16,509 unigenes were annotated by GO database (Appendix A) and 6418 unigenes by the KEGG database (Appendix A).

### 2.2. Identification and Verificationof DEGs in Transcriptome

In total, 18,826 genes exhibited significant differences in expressions between Par-J3/J4 and Pre-J2 (|log2FoldChange| > 1, padj < 0.05), including10,552 down-regulated DEGs and 8274 up-regulated DEGs in Par-J3/J4 (Figure 2a,b). From transcriptome DEGs of *M. javanica*, eight unigenes were selected for RT-qPCR to evaluate the accuracy of the transcriptome data. Four of these had high FPKM values at Par-J3/J4, while the other four had high FPKM values at Pre-J2., The relative expression at Par-J3/J4 was obtained based on that the relative expression at Pre-J2 was assigned as 1 (Figure 2c). It can be seen that the trend of relative expression is consistent with the trend of FPKM value.

Furthermore, the DEGs were classified according to GO and KEGG terms. In detail, the 10,552 down-regulated DEGs were enriched in 30 GO terms including molecular function (MF, 9251), cellular component (CC, 1217) and biological process (BP, 8426). In the 30 GO terms, the DEGs mainly responded to substrate-specific channel activity in MF, troponin complex in CC and G-protein receptor signaling path ways in BP (Figure 3a). Moreover, the 10,552 down-regulated DEGs were classified into 20 KEGG pathways. It was found that these DEGs showed close associations with metabolism, longevity, proliferation and calcium signaling pathways (Figure 3b). In a similar way, the 8274 up-regulated DEGs were enriched in 30 GO terms, and the largest number of unigenes were related to structural molecule activity in MF, sibosome in CC and the organonitrogen compound biosynthetic process in BP (Figure 3c). In addition, the 8274 up-regulated DEGs were classified into 20 KEGG pathways (Figure 3d), and these DEGs showed close associations to ribosome, oxidative phosphorylation, DNA replication and fatty acid metabolism.

### 2.3. Screening of Lectins and Sequence Alignment of the Mj-CTL Genes

DEGs with homologies to lectin domains were screened and studied [10]. There were 53 lectin proteins consisting of seven types of lectin domains, including the CTLD (**10**), galectin domain (**27**), calreticulin domain (**1**), legume lectin-like domain (**3**), LysM domain (**3**), RTB lectin domain (**6**), and the GH18 domain of class V chitinase (**3**) (Appendix A). There are 10 CTLs in *M. javanica*, 48 in *M. graminicola* and 33 in *Hirschmanniella oryzae*, respectively, but they were not found in *P. thornei*. In addition, there are 27 galectins, accounting for the majority, three chitinases and one calreticulin in *M. javanica*, and they could be also detected in another six species of plant-parasitic nematodes. In addition, six RTB lectins could be detected in *M. javanica*, but not in the soybean cyst nematode, and legume lectins have only been found in *M. javanica*, *H. oryzae* and *H. avenae*.

Among the 10 CTL genes of *M. javanica*, 4 were highly expressed at Pre-J2, and 3 had the N-terminal signal peptides, named *Mj-CTL-1*, *Mj-CTL-2* and *Mj-CTL-3.* The amino acid sequences of *Mj-CTLs* have four conserved “C” (cysteine) sites, one conserved “WIGL” domain and one conserved “WND” motif (Figure 4).

### 2.4. Suppression of Gpa2/RBP-1-Induced Cell Death by Mj-CTL-2

Previous studies have confirmed that CTLs from plant-parasitic nematodes can promote parasitism by suppressing plant defenses [12,16], so the possible role of the CTLs in *M. javanica* in suppression of host defense was investigated herein. The results of the inhibition of cell death experiment revealed that *Mj-CTL-2* exerted a striking inhibitory effect on cell necrosis (cell necrosis rate was 18%), while *Mj-CTL-1* and *Mj-CTL-3* did not markedly suppress cell necrosis (cell necrosis rates were 67% and 60%, respectively) (Figure 5b).

### 2.5. Spatial and Developmental Expression Analysis of Mj-CTL-2

As we have demonstrated that *Mj-CTL-2* can inhibit the Gpa2/RBP-1-induced tobacco cell death, so *Mj-CTL-2* was selected for further experiments. The tissue localization of *Mj-CTL-2* was determined by in situ hybridization. As the negative control treatment, no signal was observed in sense RNA probes, while a strong signal was observed within the subventral gland cells at Pre-J2 after hybridization with the digoxigenin (DIG)-labeled antisense RNA probes (Figure 6a).

With *β-actin* as the reference gene, a RT-qPCR assay was performed to examine the expression of *Mj-CTL-2* in seven developmental time points of *M. javanica*. Using the expression level at 10 dpi as reference, the time-specific fold change difference in expression of *Mj-CTL-2* is shown in Figure 6b. It can be observed that the highest expression level of *Mj-CTL-2* appeared at Pre-J2, after which its expression level gradually declined.

### 2.6. Inhibitory Effect of Mj-CTL-2 on the ROS Burst

The plants and pathogens evolved a relationship of mutual slaughter. The immune system was activated when pathogens attacked the host with a key feature of the burst of reactive oxygen species (ROS) [17]; however pathogens usually secreted effectors to suppress the burst of ROS and developed the battle relationship between the plant and pathogen [18]. Additionally, pES vectors expressing *Mj-CTL-2* and enhanced GFP (eGFP) were introduced into tobacco leaves through agroinfiltration. At 2 days after infiltration, leaf discs were collected and exposed to flg22. The results unveiled that over-expression of Mj-CTL-2 in the plants reduced the flg22-induced ROS production in comparison with the negative control eGFP (Figure 7).

### 2.7. Attenuating Effect of in Planta RNA Interference (RNAi) of Mj-CTL-2 on Nematode Parasitism

In order to conform the effect of *Mj-CTL-2* in nematode parasitism, *tobacco rattle virus* (TRV)-mediated gene silencing was performed to silence the target gene during infection. With the nematode *β-actin* gene as a reference gene, RT-Qpcr analysis results manifested that the transcript level of *Mj-CTL-2* in nematodes at 5 dpi infesting the Ptrv2-*Mj-CTL-2*-infiltrated plants displayed a drastic reduction compared with that in the control plants (Figure 8a), demonstrating the effective gene silencing mediated through *in planta* RNAi. Other *Mj-CTL* isoforms were used to verify the specificity of the *Mj-CTL-2*-targeting RNAi by Qrt-PCR analysis. The results showed that the transcriptional expressions of these *Mj-CTL* isoforms were not affected by the *Mj-CTL-2*-targeting RNAi treatment (Figure 8a).

The pTRV2-*Mj-CTL-2*-infiltrated tomato plants had 38% fewer female nematodes than the vector Ptrv-infiltrated control plants at 30 dpi (Figure 8b). These findings suggest that *Mj-CTL-2* plays a role in nematode parasitism.

### 2.8. Interactions between Mj-CTL-2 and Solanum Lycopersicum Catalase: SlCAT2

Previous research has shown that *MiCTL1a* is able to interact with the *Arabidopsis* catalase *AtCAT3*, and the co-expression of the GFP-*MiCTL1a* and mCherry-tagged fusion of *AtCAT3* in *N. benthamiana* cells has shown that *MiCTL1a* and *AtCAT3* are co-localized mainly in the plasma membrane and seldom in the peroxisome [16]. Therefore, in this study, three catalases homologous to *AtCAT3* were obtained from tomatoes by sequence alignment (Appendix A). Then, the interactions between three candidate proteins *SlCAT1*, *SlCAT2* and *SlCAT3* and *Mj-CTL-2* were further examined using the yeast two-hybrid (Y2H) co-transformation assay. The results showed that yeast co-expressing *Mj-CTL-2* and *SlCAT2* grew on a quadruple dropout medium lacking adenine, histidine, leucine and tryptophan (Figure 9a).

The transient expression experiment of tobacco illustrated that the fluorescence signal of *Mj-CTL-2*^Δsp^ (without signal peptides) was concentrated intracellularly (Figure 9b). Furthermore, the bimolecular fluorescence complementation (BiFC) was performed to confirm the interaction between *Mj-CTL-2* and *SlCAT2* in tobacco leaves, with pES-YFPN+pES-YFPC as the empty vector serving as negative control. No fluorescence signal could be observed in the negative control, but green fluorescence signals could be observed intracellularly and around the tobacco cells in *Mj-CTL-2*^Δsp^ + *SlCAT2* treatment (Figure 9b), indicating that *Mj-CTL-2* can interact with *SlCAT2* intracellularly.

## 3. Discussion

Transcriptome sequencing has been widely used to search for effectors in plant-parasitic nematodes. For example, transcriptome sequencing of *Aphelenchoides besseyi* at mixed ages was performed in 2014, yielding 13 putative effectors specific to *A. besseyi* [19]. In 2016, three potential effector proteins were identified from the transcriptome data of *M. enterolobii*, which may inhibit the plant immune response to promote the pathogenicity of nematodes [6]. In this study, the transcriptome of *M. javanica* at Par-J3/J4 and Pre-J2 were sequenced for the first time, by which 34,047 and 45,352 unigenes were obtained, respectively. This is also the first study where the transcriptome of *M. javanica* has been compared between the pre- and post-infection period. In other plant-parasitic nematodes, some comparison data of nematode transcriptomes at different developmental stages have been obtained. In the study of Huang et al., 11,443 DEGs were obtained among eggs, juveniles, females and males of *Radopholus similis*. Among the 11,443 DEGs, 2613 were up-regulated in juveniles, which were mainly related to immunity, digestion and infection, while 3546 were down-regulated, showing associations with the metabolism, growth, proliferation, transcription and protein synthesis [20]. Zhou et al. investigated RKNs invasion and development in rice roots through RNA-seq transcriptome analysis [21]. It was found that 952 and 647 genes were differentially expressed at 6 dpi (invasion stage) and 18 dpi (development stage), respectively. Gene annotation showed that the DEGs were categorized into diverse metabolic genes and stress response genes.

Herein, the comparative transcriptome data of *M. javanica* between Par-J3/J4 and Per-J2 revealed that 18,826 unigenes displayed significantly different expressions (*p* < 0.05), among which 10,552 were down-regulated and 8274 were up-regulated in Par-J3/J4. The down-regulation unigenes were mainly related to metabolism, longevity, proliferation and calcium signalling, and these processes are associated with aging [22]. There is ample evidence that the dysregulation of calcium signaling is one of the key events in neurodegenerative processes [23]. The up-regulation of unigenes was mainly associated with ribosomes, oxidative phosphorylation, DNA replication and fatty acid metabolism, etc. The changes in the expression levels of these genes are in line with the developmental progression of *M. javanica* from Per-J2 to Par-J3/J4.

Among the DEGs of *M. javanica*, 53 lectin genes with seven types of lectin domains were filtered from the transcriptome data, including the CTLD (**10**), galectin domain (**27**), calreticulin domain (**1**), legume lectin-like domain (**3**), LysM domain (**3**), RTB lectin domain (**6**), and the GH18 domain of class V chitinase (**3**). Hepatic asialoglycoprotein receptor (ASGPR) is the first identified animal CTL, and since then more than 1000 CTLs have been identified [24]. A large number of CTL genes have been identified in free-living nematodes, but few in parasitic nematodes. Loukas et al. found that CTLs secreted by animal-parasitic nematodes play a key role in host immunity [25]. Harcus et al. found three CTL genes, namely *Hp-CTL-1*, *Nb-CTL-1* and *Nb-CTL-2* in *Heligmosomoides polygyrus* and *Nippostrongylus brasiliensi* [26]. In *M. chitwoodi*, a CTL gene is secreted in the subventral esophageal gland of the nematode [27]. *Mg01965*, a CTL from *M. graminicola*, inhibits plant defense and promotes the parasitism of nematodes [12]. *MiCTL1a*, a CTL from *M. incognita*, interacts with *A. thaliana* catalase to inhibit flg22-stimulated ROS burst [16]. In this study, *Mj-CTLs* had four conserved “C” (cysteine) sites, one conserved “WIGL” domain and one conserved “WND” motif. They contain an N-terminal signal peptide but no transmembrane domain. One of the best-known characteristics of CTLDs is the “WIGL” motif, which is highly conserved in CTLDs. The “WIGL” motif is involved in the formation of hydrophobic cores in the tertiary structure of the CTL fold [10]. Moreover, the conserved “WND” motif used for calcium binding is also present in *Mj-CTLs* [28]. CTLs are produced as transmembrane proteins or secreted as soluble proteins in the parasitism [28]. The N-terminal signal peptide of *Mj-CTLs* usually aids the translocation of proteins to the endoplasmic reticulum and secretion into host plants [12].

The subventral glands produce secreted effectors of plant parasitic nematodes (PPNs) that are active during nematode penetration and at the early infection stages in roots [29]. According to a previous study, there are four effector proteins containing CTLDs, including two CTL-like proteins from *M. chitwoodi*, *Mg01965* from *M. graminicola* and *MiCTL1* from *M. incognita*, which are expressed in the subventral glands of RKNs at Pre-J2 [12,16,25]. The results of in situ hybridization showed that *Mj-CTL-2* was localized in the subventral esophageal gland cells, and its expression level was the highest at Pre-J2. It has been previously demonstrated that *Mg01965* and *MiCTL1a* secreted from nematodes into the apoplasts suppress ROS burst *in planta* [12,16]. It was found in this study that *Mj-CTL-2* had significant inhibitory effects on Gpa2/RBP-1-induced cell death and flg22-stimulated ROS burst. Moreover, *MiCTL1a* from *M. incognita* has been shown to interact with *A. thaliana* catalases: *AtCAT1*, *AtCAT2* and *AtCAT3* [16]. Herein, yeast co-transformation and BiFC assays showed that Mj-CTL-2 only interacted with *SlCAT2* intracellularly, probably because the conservation of CAT proteins in *S. lycopersicum* is lower than that in *A. thaliana* (Appendix A). *Mj-CTL-2* was silenced using an *in planta* RNAi assay, which led to significantly fewer nematodes in the roots compared to those in plants infiltrated with the pTRV vector (Figure 8b). This result, combined with the previous findings that the expression of *Mj-CTL-2* was the highest at Pre-J2, indicated that *Mj-CTL-2* indeed plays an important role in the early interaction between *M. javanica* and host plants.

In summary, 48,698 unigenes were obtained from the comparative transcriptomic analysis of *M. javanica* between Pre-J2 and Par-J3/J4. *Mj-CTL-2* is a novel CTLD-containing an effector obtained from transcriptome data. It was proposed in this study that *Mj-CTL-2* is released at the early infection stages of *M. javanica*, and that *Mj-CTL-2* interacts with *SlCAT2* to affect host defense responses by disturbing the balance of the peroxide system. CTLs may play a similar role in different plant-parasitic nematodes. The transcriptional data provide details on DEGs between Pre-J2s and Par-J3/J4. In future, we can explore more interesting effectors in the parasitic stage of *M. javanica*, as there are fewer known key effectors in regard to *M. javanica*, and it will be helpful to illustrate the molecular pathogenesis of this nematode.

## 4. Materials and Methods

### 4.1. Nematode Culture

*M. javanica* individuals were inoculated on the roots of the Xiahong No.1 tomato cultivar at 25 °C with a photoperiod of 16 h light/8 h dark in the green house at Zengcheng campus teaching & research base of South China Agricultural University as described previously [30]. Freshly hatched Pre-J2s were collected from eggs picked on the tomato root after inoculation at 30 days. Par-J3/J4s were dissected from the root after inoculation at 18 days after inoculation. Then, they were stored in a 1.5 mL centrifuge tube at −80 °C for further use.

### 4.2. RNA Extraction and Library Preparation

The RNA was extracted from approximately 20,000 Pre-J2s and 3000 Par-J3/J4s using the RNAprep pure Tissue Kit (Tiangen, Beijing, China). Sequencing libraries were constructed using the NEBNext Ultra RNA Library Prep Kit (NEB). Library fragments were purified with the AMPure XP system (Beckman Coulter, Shanghai, China) to select complementary DNA (cDNA) fragments with a preferential length of 250–300 bp. Index-coded samples were clustered on a cBot Cluster Generation System using TruSeq PE Cluster Kit v3-cBot-HS (Illumina, San Diego, CA, USA). In addition, 125 bp/150 bp paired-end reads were generated on the Illumina Hiseq platform.

### 4.3. Reads Mapping to the Reference Genome and Functional Classification

Clean reads were obtained by removing reads containing adapter, ploy-N and low-quality reads from raw data. Meanwhile, the Q20, Q30 and GC contents of the clean data were calculated. All the downstream analyses were based on the clean data with high quality. Reference genome and gene model annotation files were downloaded from the WormBase ParaSite genome database directly (https://parasite.wormbase.org/Meloidogyne_incognita_prjeb8714 (accessed on 28 May 2019)). The index of the reference genome was built using Hisat2 v2.0.5, and paired-end clean reads were aligned to the reference genome using Hisat2 v2.0.5. For function annotation, Gene Ontology (GO) analysis was performed with Blast2GO, and WEGO (https://wego.genomics.cn/ (accessed on 1 November 2018)) was used for GO term classification. Additionally, KEGG pathways were automatically generated by the KEGG Automatic Annotation Server with the BBH method (https://www.genome.jp/tools/kaas/ (accessed on 3 April 2015)).

### 4.4. Quantification of Gene Expression Level and Differential Expression Analysis

FeatureCounts v1.5.0-p3 was employed to count the read mapping to each gene, and the FPKM of each gene was then calculated based on the length of the gene and the count of reads mapped to that gene. Differential expression analysis of two conditions/groups was performed using the DESeq2 R package (1.16.1). DESeq2 provided statistical routines for determining differential expression in digital gene expression data using a model based on the negative binomial distribution. The resulting *p*-values were adjusted using the Benjamini and Hochberg approach to controlling false discovery rates. Genes with|log2FoldChange| > 1 and padj < 0.05 found by DESeq2 were assigned as differentially expressed.

### 4.5. Validation of DEGs by qPCR

Eight DEGs were selected for RT-qPCR evaluation to evaluate the accuracy of the transcriptome data obtained by Illumina sequencing. The cDNA at the two life stages was reversely transcribed using One-Step gDNA Removal and cDNA Synthesis SuperMix (TransGen Biotech, Beijing, China). Then, qPCR was performed using Two-Step RT-PCR SuperMix (TransGen Biotech) and a Takara qPCR instrument. With *Mj-actin* as a reference gene, the relative fold change was calculated using the 2^−ΔΔCt^ method [31]. Each experiment was conducted in triplicate, with three biological replicates each.

### 4.6. In Situ Hybridization

*In situ* hybridization was performed as previously described by De Boer et al. [32]. First, specific 200 bp–300 bp templates were amplified from *Mj-CTL-2*, and then DIG-labeled sense and antisense RNA probes were generated using the T7 enzyme (Promega, Madison, WI, USA). Later, the nematode sections were hybridized and examined under the ECLIPSE Ni microscope (Nikon, Tokyo, Japan). Primers used in this study were listed in Appendix A.

### 4.7. Developmental Expression Analysis

An RNA prep micro kit (Tiangen, Beijing, China) was utilized to extract the RNA from *M. javanica* at different developmental time points, including Pre-J2, 2 dpi and 5 dpi (Par-J2), parasitic third-stage juveniles at 10 dpi and 14 dpi (Par-J3), parasitic fourth-stage juveniles at 18 dpi (Par-J4), and adult female nematodes at 30 dpi. Approximately 100 nematodes at Pre-J2, Par-J3, Par-J4 and female stages were collected for RNA extraction. Regarding the Par-J2, 100 nematodes were inoculated into the tomato root, and the infected roots were collected for RNA extraction. Next, the cDNA at seven life stages was subjected to reverse transcription using One-Step gDNA Removal and cDNA Synthesis SuperMix (TransGen Biotech), followed by qPCR using Two-Step RT-PCR SuperMix (TransGen Biotech) and a Takara qPCR instrument. With *Mj-actin* as a reference gene, the relative transcript abundance was calculated using the 2^−ΔΔCt^ method [29]. Each experiment was conducted in triplicate, with three biological replicates each.

### 4.8. Inhibitory Effect of CTLs of M. javanica on Gpa2/RBP-1-Induced Tobacco Cell Death

*Nicotiana benthamiana* plants were grown at 25 °C for 4 weeks in a greenhouse with a 16 h light/8 h dark cycle. Then, *Mj-CTL-1*, *Mj-CTL-2* and *Mj-CTL-3* fragments without signal peptides were cloned into the pES vector with an eGFP-tag fused at the C-terminus to generate pES:CTL. After that, the constructs pCAMBIA1305: Gpa2 and pCAMBIA1305:Rbp-1:HA were kept in the laboratory, and experiments were conducted in line with the procedures elaborated by Chen et al. [33]. Briefly, the plasmids pES:CTL, pCAMBIA1305: Gpa2, pCAMBIA1305:Rbp-1:HA were introduced into the *Agrobacterium tumefaciens* strain EHA105. The transformed bacteria were cultured and suspended in a buffer containing 10 mM 2-(*N*-morpholino) ethanesulfonic acid (MES) (pH5.5) and 200 μM acetosyringone at a final optical density at 600 nm (OD600) of 0.3, then infiltrated into tabacco leaves with individual MES as the control. Symptoms were photographed 3 days after the last infiltration, and the suppression degree of cell death was analyzed by the necrosis rate.

### 4.9. Detection of ROS Burst

To detect the ROS generation after flg22 treatment, *Mj-CTL-2* fragments without signal peptides were cloned into the pES vector with an eGFP-tag fused at the C-terminus to generate pES: *Mj-CTL-2*^ΔSP^. Next, recombinant plasmids were transformed into the *A. tumefaciens* strain GV3101. *N. benthamiana* leaves at 4 weeks old were infiltrated with *A. tumefaciens* carrying the recombinant plasmid. After 48 h of infiltration, leaf discs were collected and put into 96-well plates (Costar 96-well white flat bottom polystyrene) containing 200 μL of double-distilled water (ddH_2_O). A luminol-based assay was used to detect ROS. After 16 h, the water was removed and 100 μL of ddH_2_O containing 34 μg luminol (Sigma, St. Louis, MO, USA), 20 μg horseradish peroxidase (Sigma) and 100 nM flg22 were added, the data were read using a Tecan Infinite 200 Pro plate reader [34].

### 4.10. In Planta RNAi

A fragment of about 300 bp of *Mj-CTL-2* was amplified by PCR. Next, the fragment was digested using XbaI and SacI, then cloned into the pTRV2 vector digested with the same enzymes to generate pTRV2. The vectors pTRV2 and the pTRV2 derived pTRV2-*CTL-2* were transformed into the *A. tumefaciens* strain EHA105, respectively. Later, tomato plants were infected by EHA105 carrying the corresponding constructs, as he procedures previously described [35]. The coat protein gene of TRV was used to verify whether viruses successfully invade cells, and it was detected using the primer pair TRVcpF/TRVcpR at 14 dpi [36]. After 21 days, 200 freshly hatched nematodes were successfully inoculated into the roots of plants expressing TRV. At 5 dpi, the RNA was extracted from plant roots, and RT-qPCR was performed to detect the silencing efficiency of target genes in nematode. Independent RT-qPCR experiments were performed three times. After 30 days, the roots were taken out and stained with sodium hypochlorite and acid fuchsin. Finally, nematodes in the roots were counted.

### 4.11. Y2H Co-Transformation and BiFC Assays

The encoding sequence of *Mj-CTL-2* without signal peptide was cloned into the pGBKT7 vector to generate pGBKT7: *Mj-CTL-2*^ΔSP^ as the prey vector. The open reading frames (ORFs) of *SlCAT1*, *SlCAT2* and *SlCAT3* were inserted into the pGADT7 vector to generate pGADT7: *SlCAT1*, pGADT7: *SlCAT2* and pGADT7: *SlCAT3*, respectively, as the bait vector. Then, the prey and bait plasmids were co-transformed into the yeast strain Y2HGOLD, and the transformants were grown on an SD/LeuTrp selection medium. Finally, positive clones were selected and cultured on the SD/LeuTrpHisAde medium [37].

*Mj-CTL-2* with or without signal peptides was cloned into the pES-YFPN vector to generate pES-YFPN: *Mj-CTL-2* and pES-YFPN: *Mj-CTL-2*^ΔSP^. The ORFs of *SlCAT1*, *SlCAT2* and *SlCAT3* were cloned into the pES-YFPC to generate pES-YFPC: *SlCAT2*. Then, the recombinant plasmids were transformed into the *A. tumefaciens* strain GV3101. *N. benthamiana* leaves at 4 weeks old were co-infiltrated with *A. tumefaciens* carrying the plasmids pES-YFPN: *Mj-CTL-2* and pES-YFPC: *SlCAT2*, pES-YFPN: *Mj-CTL-2*^ΔSP^ and pES-YFPC: *SlCAT2* at a 1:1 ratio. After 48 h of infiltration, leaves were collected and observed under a confocal microscope (Nikon, Tokyo, Japan).

## Figures and Tables

**Figure 1 plants-13-00730-f001:**
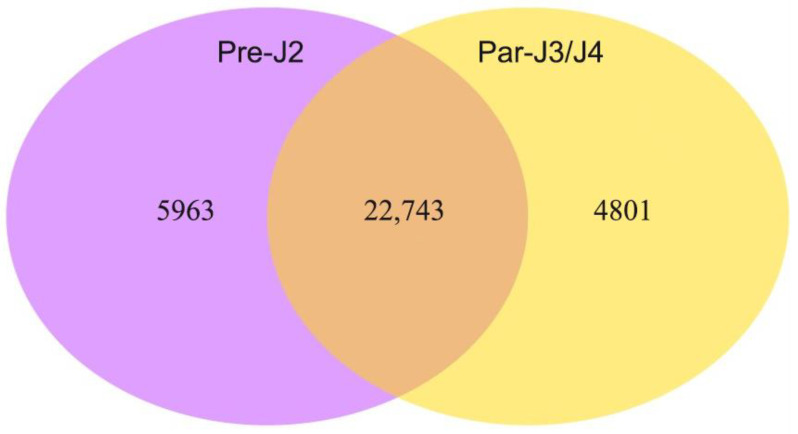
Venn diagram of DEGs between Pre-J2 and Par-J3/J4.

**Figure 2 plants-13-00730-f002:**
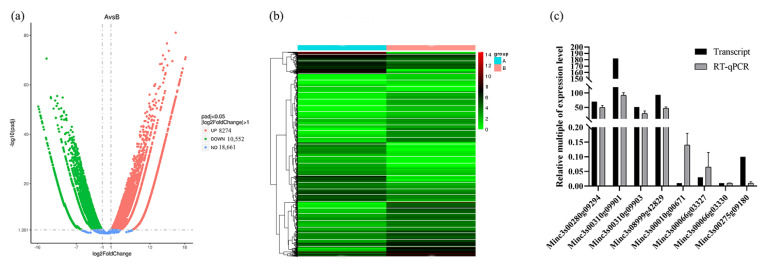
The analysis and verification of DEGs between Par-J3/J4 and Pre-J2. (**a**) Volcano map of DEGs in the transcriptomes at Par-J3/J4 compared to Pre-J2. Red dots indicate up-regulated DEGs, green dots indicate down-regulated DEGs, and blue dots indicate unchanged genes in Par-J3/J4 (A) relative to Pre-J2 (B). The *x*-axis shows log2FoldChange, the *y*-axis represents -og10(padj), padj: *p*-value after correction for multiple hypothesis testing. (**b**) Heatmap of DEGs at Par-J3/J4 (A) and Pre-J2 (B). Left half indicates the FPKM of DEGs at Par-J3/J4 (A), whereas the right half indicates the FPKM of DEGs at Pre-J2 (B). (**c**) RT-qPCR verification of 8 differentially expressed genes, β-actin was used as the reference gene.

**Figure 3 plants-13-00730-f003:**
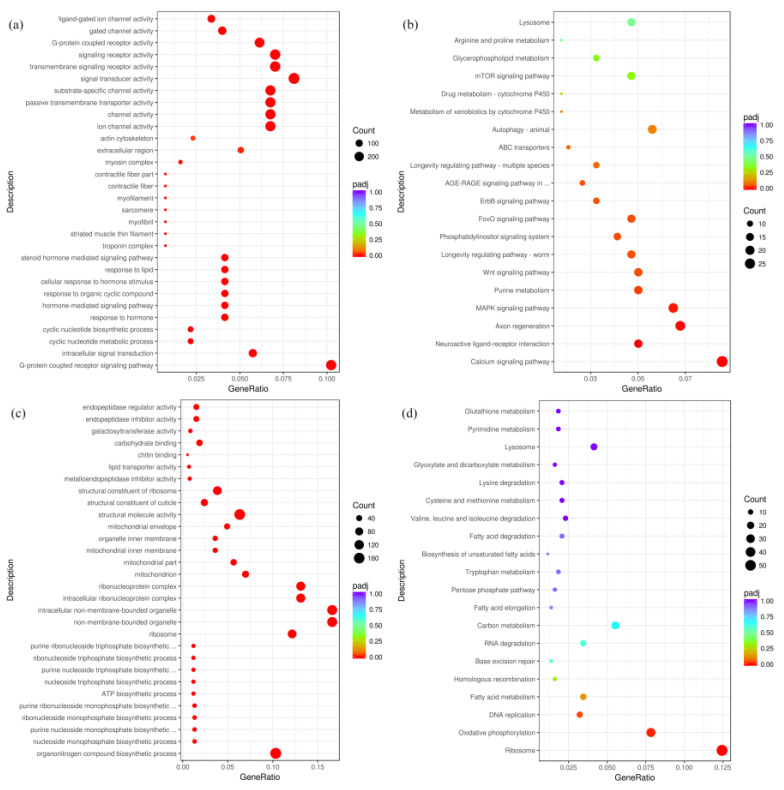
The top 30 enriched GO terms of down-regulated DEGs and up-regulated DEGs, respectively (**a**,**c**). Twenty KEGG pathways of the down-regulated DEGs and up-regulated DEGs, respectively (**b**,**d**).

**Figure 4 plants-13-00730-f004:**
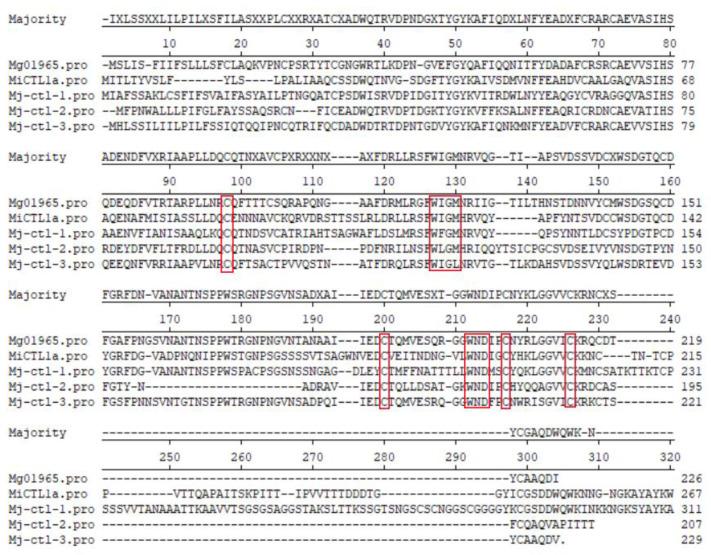
Amino acids alignment of CTLs from *M. javanica*, *M. graminicola*, and *M. incognita*. The red box reflects the position of the conservative amino acid.

**Figure 5 plants-13-00730-f005:**
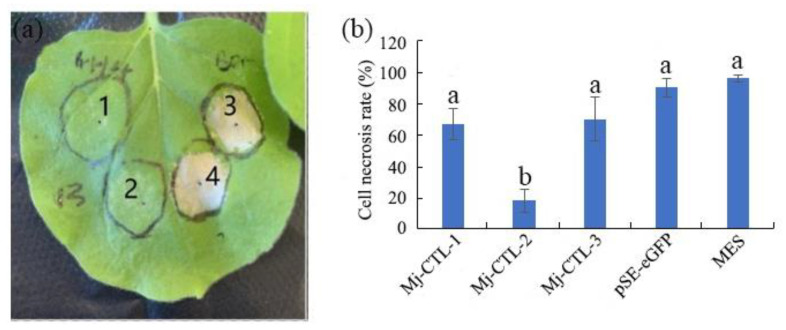
Mj-CTL Suppress the cell death triggered by Gpa2/RBP-1 (**a**) Cell death in tobacco leaves. 1: Mj-CTL-2; 2: Mj-CTL-2 → 24 h → Gpa2/RBP-1; 3: PES-GFP → 24 h → Gpa2/RBP-1; 4: MES → 24 h → Gpa2/RBP-1. (**b**) Percentage of dead cells. The different lowercase letters indicate statistically significant differences based on ANOVA with the Duncan post hoc test (*p* < 0.05).

**Figure 6 plants-13-00730-f006:**
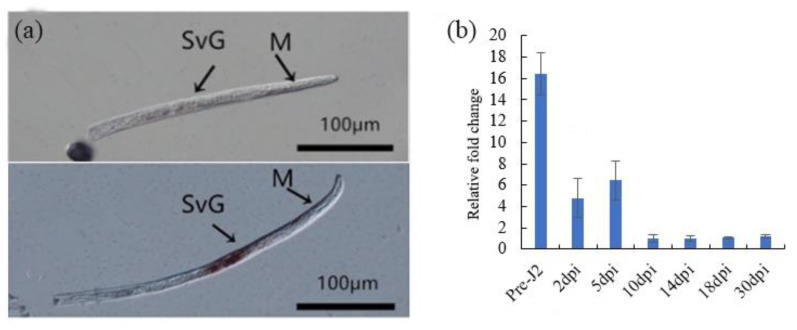
Expression patterns of Mj-CTL-2 in *M. javanica*. (**a**) In situ hybridization of Mj-CTL-2. Fixed nematodes are hybridized with sense (above panel) and antisense mRNA probes (below panel) from Mj-CTL-2. SvG: subventral esophageal gland cells; M: metacorpus. Bars, 100 μm. (**b**) The expression pattern of Mj-CTL-2. With β-actin as the reference gene, the stage-specific expression of Mj-CTL-2 is detected by qRT-PCR at seven different life stages of *M. javanica*. The *x*-axis represents seven stages, while the *y*-axis represents relative expression. The fold change values were calculated using the 2^−ΔΔCT^ method and presented as the change in the mRNA level at various time points after inoculation. Each column represents the means of three independent experiments with standard deviation.

**Figure 7 plants-13-00730-f007:**
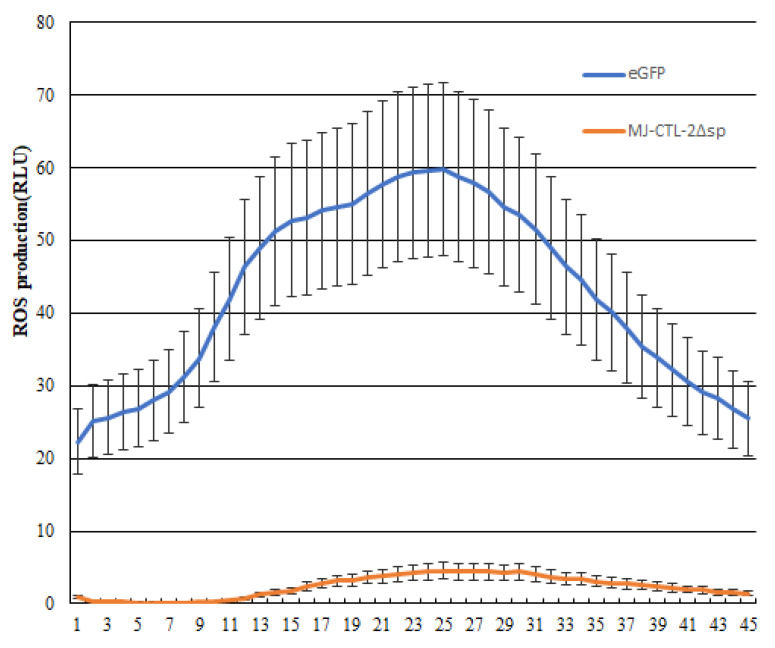
Mj-CTL-2 suppresses flg22-mediated ROS production in *N. benthamiana*. Values indicated are the average relative luminescence units (RLUs) from two leaf disc replicates per leaf, with eGFP: enhanced green fluorescent protein, and Mj-CTL-2Δsp: Mj-CTL-2 without signal peptides.

**Figure 8 plants-13-00730-f008:**
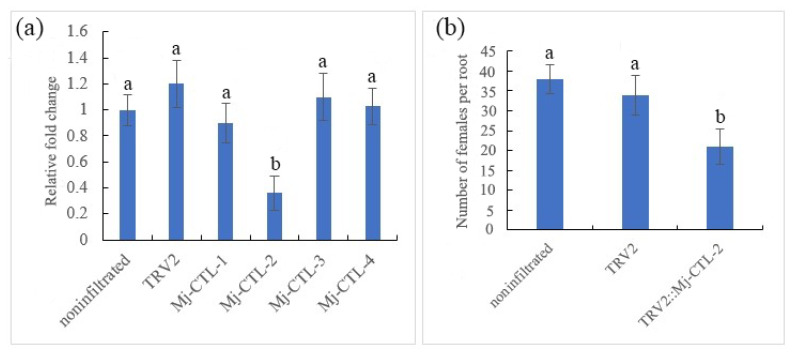
Effect of in planta RNAi of Mj-CTL-2 on *M. javanica* parasitism. (**a**) The qRT-PCR assays of the expression levels of Mj-CTL-2 in *M. javanica* collected from non-infiltrated tomato plants and pTRV2, pTRV2-Mj-CTL-2 agroinfiltrated plants. The expression levels of Mj-CTL isoforms from *M. javanica* were quantified to determine the specificity of RNAi. (**b**) The number of adult females per root. The different lowercase letters indicate statistically significant differences based on ANOVA with the Duncan post hoc test (*p* < 0.05). The experiment was performed with three biological repeats.

**Figure 9 plants-13-00730-f009:**
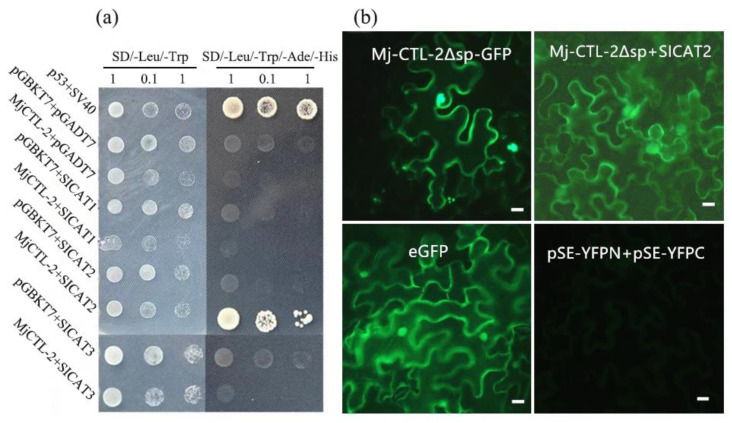
Mj-CTL-2 interacts with *Solanum lycopersicum* catalase: SlCAT. (**a**) Yeast two-hybrid tests between Mj-CTL-2 and SlCATs. Left column: yeast cell growth carrying the baits (in pGBKT7 vector) and preys (in pGADT7) grown on SD/LeuTrp medium. Right column: yeast cell growth on the selective quadruple dropout medium SD/LeuTrpAdeHis. Yeast cells containing p53 and SV40 large T-antigen are used as the positive control, and those containing pGBKT7 and pGADT7 are used as the negative control. (**b**) BiFC validation of the interaction between Mj-CTL-2 and SlCAT2 in *N. benthamiana* leaves. The fluorescence signal is detected at 48 h after infiltration. Images are captured by confocal microscopy. Bars, 20 μm.

## Data Availability

Data are contained within the article and Appendix A.

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
