# Peer review of "Transcriptome Analysis of Meloidogyne javanica and the Role of a C-Type Lectin in Parasitism"

_plants, 2024, doi:10.3390/plants13050730_

Round 1
Reviewer 1 Report
Comments and Suggestions for Authors
Reviewer 2 Report
Comments and Suggestions for Authors
In this comprehensive study by Chi et al., a comparative transcriptome analysis of Meloidogyne javanica was performed between two life stages, pre and post host plant infestation. This was followed up with many informative assays regarding particularly interesting and likely biologically relevant candidates from the transcriptome analysis. The amount of work that went into this study is truly impressive and I find the authors have done a good job of producing reliable and informative results that help in our understanding of the parasitic nature of these root-knot nematodes and how they suppress plant defence. Although the overall flow of the article is well structured, there are areas where the rational for specific experiments seem to be lacking or only added later through the introduction of another assay interrogating a similar set of hypotheses (for example results section 2.6 could use a sentence explaining the why assessing ROS is significant, as its significance in plant perception of attack was not mentioned in the introduction). The results section could also use some clean up with regards to formatting of plots, namely that much of the text is completely unreadable unless you magnify 200x (Figure 3, supplemental Figure 1). The figures sometimes have outlines they shouldn't have (example Figure 6b). The text colour appears to be different in some cases (example Figure 5b with grey axes and black statistical comparison letters). The discussion section is well elaborated upon, and the results are well placed in the context of previously performed studies. It might be nice to know what the authors think the next steps should be in terms of further interrogating how M. javanica uses their plant hosts, or perhaps more uses for the transcriptional data already generated. The methods section could use some more information regarding the specifics of how experiments were done and rely less on citations of other work, as they are not always useful (lead to other articles that cite other articles) and makes it difficult for the reader to focus on this article and results in context. A brief description of the method is always required even if a more detailed description is cited for the reader to look up if they are interested in such details. In general, this was a well performed set of experiments which were appropriately analysed and interpreted as far as this reviewer can tell, not being an expert in nematodes. After the introduction of more information, rationales and clearing up some inconsistencies between the results and methods section, I believe this article should be published in Plants. My suggested edits and more detailed comments can be found in the accompanying PDF.

There were no detected issues with the English, only minor edits required.
Author Response
Response to Reviewer X Comments
|
||
1. Summary |
|
|
Thank you very much for taking the time to review this manuscript. Please find the detailed responses below and the corresponding revisions highlighted with red letters in the re-submitted files. |
||
2. Questions for General Evaluation |
Reviewer’s Evaluation |
Response and Revisions |
Does the introduction provide sufficient background and include all relevant references? |
No objections |
|
Are all the cited references relevant to the research? |
No objections |
|
Is the research design appropriate? |
No objections |
|
Are the methods adequately described? |
No objections |
|
Are the results clearly presented? |
In this comprehensive study by Chi et al., a comparative transcriptome analysis of Meloidogyne javanica was performed between two life stages, pre and post host plant infestation. This was followed up with many informative assays regarding particularly interesting and likely biologically relevant candidates from the transcriptome analysis. The amount of work that went into this study is truly impressive and I find the authors have done a good job of producing reliable and informative results that help in our understanding of the parasitic nature of these root-knot nematodes and how they suppress plant defence. |
Thank you very much for the suggestions, and we are very glad to receive the positive assessment of our manuscript. We checked all the suggestions and annotations in PDF, and thank you very much for the careful work. |
Are the conclusions supported by the results? |
No objections |
|
3. Point-by-point response to Comments and Suggestions for Authors |
||
Comments 1: Results section 2.6 could use a sentence explaining the why assessing ROS is significant, as its significance in plant perception of attack was not mentioned in the introduction.
|
||
Response 1: Thank you very much for the suggestions. A sentence was placed in the section 2.6 as below. Plants and pathogens evovled a relationship of mutual slaughter. The immune system was actived when pathogens attack the host with a key feature of the burst of reactive oxygen species (ROS) [17]. But pathogens usually secreted effectors to suppress the burst of ROS, and developed the battle realationship between palnt and pathogen [18].
|
||
Comments 2: The results section could also use some clean up with regards to formatting of plots, namely that much of the text is completely unreadable unless you magnify 200x (Figure 3, supplemental Figure 1). |
||
Response 2: Thank you very much for the suggestions. It is probably that the images in the text were compressed. So in the next step we will provide the original images.
|
||
Comments 3: The figures sometimes have outlines they shouldn't have (example Figure 6b). |
||
Response 3: Thank you very much for the attention. The outlines of Figure 6b and Figure7 were removed.
|
||
Comments 4: The text colour appears to be different in some cases (example Figure 5b with grey axes and black statistical comparison letters). |
||
Response 4: Thank you very much for the attention. The grey axes were revised to black.
|
||
Comments 5: The discussion section is well elaborated upon, and the results are well placed in the context of previously performed studies. It might be nice to know what the authors think the next steps should be in terms of further interrogating how M. javanica uses their plant hosts, or perhaps more uses for the transcriptional data already generated. |
||
Response 5: Thank you very much for your approval. The transcriptional data provide a detail DEGs between Pre-J2s and Par-J3/J4. In the future we can explore more interesting effector in the parasitic stage of M. javanica, bucause there are fewer key effectors are known in M. javanica. It will helpful to illustrate the molecular pathogenesis of this nematode.
|
||
Comments 6: The methods section could use some more information regarding the specifics of how experiments were done and rely less on citations of other work, as they are not always useful (lead to other articles that cite other articles) and makes it difficult for the reader to focus on this article and results in context. A brief description of the method is always required even if a more detailed description is cited for the reader to look up if they are interested in such details. |
||
Response 6: Thank you very much for the suggestions. We have added the brief description in the methods section. They are mainly in section 4.1, section 4.2, section 4.7, section 4.8, section 4.9, and section 4.11. Please refer to the main text for the details.
|
||
4. Response to Comments on the Quality of English Language |
||
Point 1: No objections |
||
5. Additional clarifications |
||
No information |

Reviewer 3 Report
Comments and Suggestions for Authors
Interesting paper where comparison of two stages (pre and post infection period) are compared at first time in Meloidogyne javanica, through transcriptomic analysis. Although parasitism role of nematode´s Lectins is well known for those of animal parasites, scarce literature have been produced related to Plant Parasitic nematodes. For special interest is the fact that the experiment is devoted to one of the most important plant parasitic nematodes such as M. javanica whose role in Phytopathology is getting increment around the World and conclussions can be extended to other Meloidogyne spp.
This work is quite well designed and performed. The authors follow a canonical thecnical and scientific methodology which facilitates comparison with other similar studies carried out in other different fields (medical, paratitology, taxonomy or ecology) and replication of experiments. In my opinion this paper deserves to be published in its present form.
Author Response
Thank you very much for the approval.